# A Fatal Case of *Rhizopus azygosporus* Pneumonia Following COVID-19

**DOI:** 10.3390/jof7030174

**Published:** 2021-02-28

**Authors:** Anubhav Kanwar, Alex Jordan, Scott Olewiler, Kurt Wehberg, Michael Cortes, Brendan R. Jackson

**Affiliations:** 1Department of Infectious Diseases and Travel Medicine, Beebe Healthcare, Lewes, DE 19958, USA; solewiler@beebehealthcare.org; 2Epidemiologist, Mycotic Diseases Branch, Centers for Disease Control and Prevention, Atlanta, GA 30329-4018, USA; noq1@cdc.gov; 3Department of Cardiothoracic Surgery, Beebe Healthcare, Lewes, DE 19958, USA; kwehberg@beebehealthcare.org; 4Department of Internal Medicine, Beebe Healthcare, Lewes, DE 19958, USA; mcortes@beebehealthcare.org; 5Late Sequelae Unit, CDC COVID-19 Response Clinical Team, Centers for Disease Control and Prevention, Atlanta, GA 30329-4018, USA

**Keywords:** mucormycosis, COVID-19, pneumonia, *Rhizopus*

## Abstract

We report a fatal case of *Rhizopus azygosporus* pneumonia in a 56-year-old man hospitalized for COVID-19 who had received methylprednisolone and tocilizumab. Although COVID-associated pulmonary aspergillosis has been widely documented, mucormycosis has been rarely reported. In this patient, receipt of two commonly used immunosuppressants likely contributed to mucormycosis risk, suggesting the need for vigilance in hospitalized patients with COVID-19.

## 1. Introduction

Mucormycosis is a life-threatening fungal infection caused by molds belonging to the subphylum Mucoromycotina in the order Mucorales [1]. Mucormycoses are most commonly seen in immunocompromised patients, including those with hematologic malignancies, transplants, and extensive corticosteroid use, although they are also reported in immunocompetent patients with diabetes mellitus, penetrating trauma, iron overload, and injection drug use [2]. Major sites of infection are sinuses, lungs, skin, brain, and gastrointestinal tract.

Invasive fungal infections have been increasingly reported in patients with coronavirus disease-2019 (COVID-19), primarily invasive candidiasis and pulmonary aspergillosis [3,4]. Aspergillosis, like mucormycosis, has classically been seen primarily in immunocompromised patients; however, it has been increasingly observed in patients admitted to intensive care units, patients with severe influenza, and now COVID-19, termed COVID-associated pulmonary aspergillosis (CAPA) [5]. These infections may reflect impaired mucosal barrier and a dysfunctional immune response in severe viral infections and the use of immunosuppressive medications like corticosteroids and interleukin (IL-6) inhibitors like tocilizumab [6,7]. Few cases of invasive mucormycosis in patients with COVID-19 have been reported. A PubMed search performed on December 26th, 2020, with search terms zygomycosis, mucormycosis, phycomycosis, *Rhizopus*, *Mucor*, *Rhizomucor*, *Cunninghamella*, *Absidia, Apophysomyces, Syncephalastrum, Saksenaea, Cokeromyces, Entomophthora, Conidiobolus*, and *Basidiobolus* identified six cases, with three cases of orbital mucormycosis, two cases of pulmonary mucormycosis, and one case of gastrointestinal mucormycosis [8,9,10,11,12,13]. In addition, there have been recent news reports describing an outbreak of mucormycosis among patients with COVID-19 in hospitals in Delhi, India [14].

## 2. Case Report

We report a case of a 56-year-old man with end-stage renal disease on hemodialysis who ultimately developed mucormycosis as a complication of COVID-19. In early November 2020, he had a positive nasopharyngeal PCR test for severe acute respiratory syndrome coronavirus-2 (SARS-CoV-2) while asymptomatic. Four days later, he was hospitalized for fatigue and shortness of breath, at which time he received a five-day course of methylprednisone, a single dose of tocilizumab, and one unit of convalescent plasma. Blood cultures collected on admission were negative for bacterial and fungal organisms, and he lacked productive cough for sputum culture. He was discharged home seven days later on 3 L/min of oxygen. 

Five days after discharge (16 days after the initial SARS-CoV-2 test), he was readmitted (hospital day (HD) 0) with generalized fatigue, shortness of breath, and hemoptysis (Figure 1). He was afebrile, with a blood pressure of 169/80 mmHg, respiratory rate of 20/min, and oxygen saturation of 99% on 2 L of oxygen. Despite continuing outpatient dialysis, his weight was 3 kg greater than his previous discharge weight of 92 kg. He had leukocytosis (11,200 cells per microliter), with lymphopenia (5.2%), and his procalcitonin was 1.36 ng/mL (reference range: <0.15 nanogram/mL). Chest radiograph showed increasing airspace density in both lungs and pleural effusion. Nasopharyngeal SARS-CoV-2 testing was again positive by PCR. He was started on empiric intravenous (IV) vancomycin and piperacillin-tazobactam for concerns of healthcare-associated pneumonia. Blood and sputum cultures obtained before antibiotic initiation were negative.

A repeat sputum sample collected on HD 3 revealed filamentous fungal elements on fungal stain, and the culture was sent to Mayo Clinic Laboratory (Rochester, Minn.) for further identification. Empiric liposomal amphotericin B (LAmB, 5 mg/kg) was started, and antibacterial medications were discontinued. Computed tomography (CT) of the chest on HD 3 showed patchy ground glass infiltrates and a moderate loculated right-sided pleural effusion with right sided cavitary pneumonia (Figure 2D). On HD 5, ultrasound-guided thoracentesis yielded cloudy-appearing fluid with 28,000 leukocytes (88% neutrophils, 12% monocytes), pH 7.7, protein 2.8 gm/dl, lactate dehydrogenase (LDH 851) units/L, and glucose 222 mg/dl; fungal stain revealed filamentous fungi. On HD 7, cotton-fluffy growth was seen on the sputum culture on Sabouraud’s dextrose agar slant, which was suspected to be fungus from the Mucorales group based on pauci-septate hyphae with unbranching sporangiophores seen on lactophenol-cotton blue stain. Similar growth was observed from a pleural fluid sample (Figure 2A,B). A pigtail catheter was placed on HD 10 to drain the persistent pleural effusion.

Repeat chest CT on HD 12 showed unchanged pleural effusion, as well as dense consolidation in the right lower lobe with lung necrosis (Figure 2E). Given the isolation of a fungus from pleural fluid and lack of improvement following catheter placement, the patient underwent robotic decortication surgery on HD 13 as a salvage procedure. Intraoperative irrigation with LAmB was done, and two chest tubes were placed for continued drainage. The filamentous fungus from sputum and pleural fluid was identified by Mayo Clinic Laboratories as *Rhizopus* spp. by MALDI-TOF (Matrix-assisted laser desorption ionization time-of-flight) with a minimum inhibitory concentration (MIC) of 0.125 µg/mL for LAmB, 0.25 µg/mL for posaconazole, and 2 µg/mL for isavuconazole, all suggestive of susceptibility. The pleural fluid isolate was subsequently sent to the Centers for Disease Control and Prevention Mycotic Diseases Branch laboratory, which identified it as *Rhizopus azygosporus* based on sequence analysis of the internal transcribed spacer (ITS) and D1/D2 ribosomal DNA regions. Histopathological examination of parietal pleura and cortical rind showed necrotic tissue and purulent exudate admixed with numerous fungal organisms morphologically consistent with mucormycetes on Grocott’s methenamine silver and periodic acid–Schiff stains (Figure 2C). On HD 15, the patient was extubated but needed reintubation the next day due to increased oxygen requirements. The patient became hypotensive to <65 mm Hg. Intravenous norepinephrine and empiric IV vancomycin and piperacillin-tazobactam were reinitiated, and blood cultures later grew vancomycin-resistant *Enterococcus* spp. and *Bacteroides fragilis*. On the following day (HD 17), the patient developed cardiac arrest and died.

## 3. Discussion

Although invasive mold infections caused by *Aspergillus* have increasingly been reported in patients with severe COVID-19, affecting as many as 14% of ICU patients in one cohort [15], we identified only six previously reported COVID-associated mucormycosis cases worldwide. The case presented here is thus one of the first reported cases of COVID-associated mucormycosis and is notable for the presence of necrotizing pneumonia and empyema. The two pulmonary COVID-associated mucormycosis cases published previously involved a 49-year-old man in Yuma, Arizona, who developed bronchopleural fistula requiring thoracotomy and fistula repair with pleurodesis and a 66-year-old man in Sassari, Italy, with pulmonary mucormycosis with extensive cavitary lesions, who was managed medically with LAmB given his high operative risk [11,12]. Neither patient had an underlying condition before COVID-19, and both died during the hospital stay. All six previously reported patients with COVID-associated mucormycosis had received broad-spectrum antibacterial medications initially and were tested for non-bacterial etiologies only after their clinical status worsened. Two of these patients with pulmonary mucormycosis received IV corticosteroids and a single dose of tocilizumab [11,12]. That all three patients with pulmonary disease, including the one presented here, died, underscores the severity of mucormycosis in the setting of severe COVID-19. Delayed recognition and treatment may have contributed to the poor outcomes. That none of the three had an underlying condition that classically predisposes to mucormycosis suggests a causative role for COVID-19, possibly compounded by the use of corticosteroids and an IL-6 inhibitor. 

Severe COVID-19 should be considered as a risk factor for invasive fungal infections, particularly in patients who receive immunosuppressive medications like corticosteroids (now standard of care for severe COVID-19) and IL-6 inhibitors. The European Society for Clinical Microbiology and International Society of Human and Animal Mycology recently published guidance to identify proven, probable, and possible CAPA, including use of radiologic findings, microbiology, histopathology, galactomannan antigen testing, and PCR [16]. Since microbiology and histopathology specimens can be difficult to obtain in patients with COVID-19 to diagnose CAPA, many cases may be diagnosed through less invasive galactomannan and *Aspergillus* PCR testing in the presence of specific clinical factors. However, neither of these diagnostics will detect mucormycete infections, and clinicians should maintain some index of suspicion for mucormycosis despite the rarity of reported cases to date. As in the case presented here, culture or histopathology may be needed for diagnosis, although molecular diagnostics such as quantitative PCR assays are also available [4].

Given limited diagnostics available for mucormycosis, an urgent need exists to develop non-invasive tests [17]. An enzyme-linked immunospot (ELISpot) assay, which detects mucormycete-specific T-cells, analogous to interferon-gamma-release assays for detection of latent tuberculosis, has been evaluated in patients with hematologic malignancies [18]. Whether this test would be useful in detecting COVID-associated mucormycosis remains to be seen. 

We hypothesize that the actual number of COVID-associated mucormycosis might be much higher than reported given the challenges in diagnosis and likely under-detection, supported by emerging reports of cases from India. These challenges might include high overall mortality in severe COVID-19 with few autopsies conducted, avoidance of aerosol-generating procedures like bronchoscopy-assisted sampling, lack of rapid fungal diagnostics, and overburdened healthcare systems with shortages of healthcare providers. The role of greater recent use of glucocorticoids for severe COVID-19 remains to be evaluated. Greater study and surveillance of invasive mold infections, not just aspergillosis but also mucormycosis, in severe COVID-19, would help determine the true burden and most effective diagnostics, treatment, and prevention.

## Figures and Tables

**Figure 1 jof-07-00174-f001:**
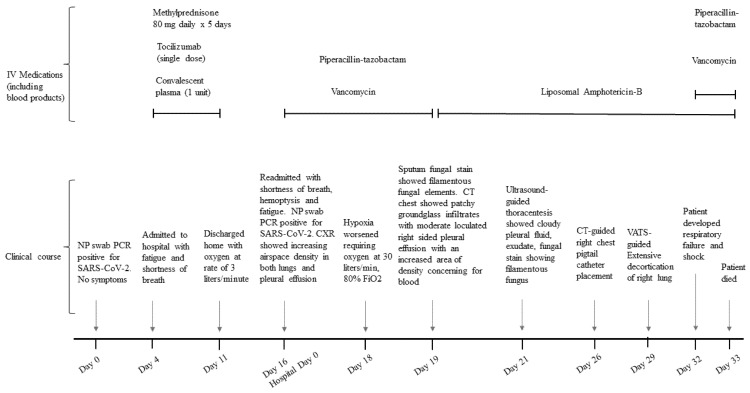
Clinical course of a patient with COVID-19 who developed pulmonary mucormycosis.

**Figure 2 jof-07-00174-f002:**
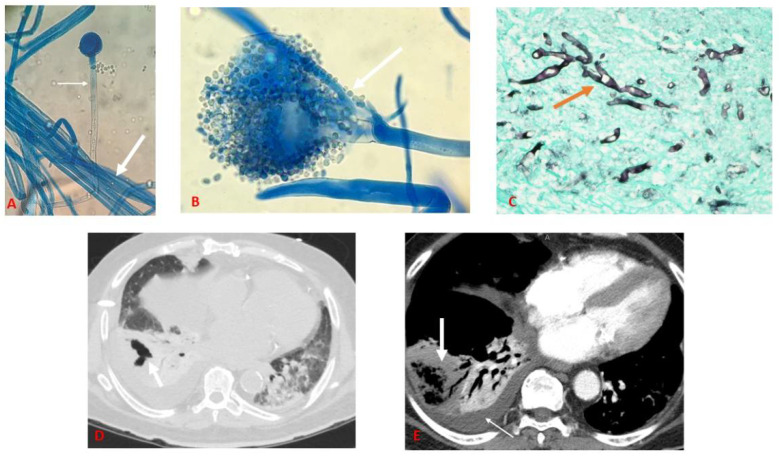
(**A**) Pleural fluid culture growing multiple pauci-septate hyphae (thick arrow) with a sporangiophore in focus at 10× (thin arrow); (**B**) Pleural fluid culture with a sporangium containing multiple round sporangiospores in 40×(thick arrow); (**C**) Pleural decortication specimen stained with GMS stain under 40× magnification showing purulent exudate admixed with fungal organisms (orange arrow); (**D**) Computed tomography of chest with axial view showing cavity (thick arrow) in lung window; (**E**) Computed tomography angiogram of chest with axial view showing lung necrosis (thick arrow) and pleural effusion (thin arrow).

## Data Availability

Not applicable.

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
