# Peer review of "A Fatal Case of Rhizopus azygosporus Pneumonia Following COVID-19"

_jof, 2021, doi:10.3390/jof7030174_

Round 1

Reviewer 1 Report

Dear authors:

Many thanks for your contribution. Here my main concerns.

Lines 78-81: Please explain in details how the preliminary fungal identification was proposed? Authors only mentioned that a Mucorales group member was presumably identified based on morphology. You need to explain this.

Lines 88-89: Why the Rhizopus strain was identified? Please, explain and show evidences about the used methodology to obtain this taxonomic identification. It is inadmissible to accept the proposal identification only with the mentioned information in the text. In addition, how did you take the sample? How many samples did you obtain? Was the procedure performed in triplicate (at least)? How did you discharge a microbial contamination during sampling? Did you only isolate ONE fungi from the clinical sample? Please, explain all these questions in detail. 

Lines 91-94: This information is vague. You have to demonstrate that fungi observed in purulent exudate as really mucormycetes. The evidences provided in the text are not enough.

Line 99: How did you obtain the final fungal identification? You propose a taxonomic identification at species level. How did you obtain it?

Author Response

Lines 78-81: Please explain in details how the preliminary fungal identification was proposed? Authors only mentioned that a Mucorales group member was presumably identified based on morphology. You need to explain this.
--- We have added additional information. Preliminary fungal identification was based on demonstration of pauci-septate hyphae with unbranching sporangiophores on lactophenol-cotton blue stain. 

Lines 88-89: Why the Rhizopus strain was identified? Please, explain and show evidences about the used methodology to obtain this taxonomic identification. It is inadmissible to accept the proposal identification only with the mentioned information in the text. In addition, how did you take the sample? How many samples did you obtain? Was the procedure performed in triplicate (at least)? How did you discharge a microbial contamination during sampling? Did you only isolate ONE fungi from the clinical sample? Please, explain all these questions in detail.

---We have added additional information about how the Rhizopus strain was identified. The sputum and pleural fluid samples were collected by the usual methods used in clinical practice, which does not include plating isolates in triplicate. We are not sure what you mean by “discharge a microbial contamination” but the fact that this same molecularly identified organism was cultured from both sputum and pleural fluid argues against contamination. The histopathology results, consistent with the molecular identification, add further support that this represented true infection. Only one fungal species was grown and identified.

Lines 91-94: This information is vague. You have to demonstrate that fungi observed in purulent exudate as really mucormycetes. The evidences provided in the text are not enough.

--- We have added additional information highlighting the use of sequence analysis of the internal transcribed spacer (ITS) and D1/D2 ribosomal DNA regions of the isolate to identify the fungi as Rhizopus azygosporus (a mucormycete)

Line 99: How did you obtain the final fungal identification? You propose a taxonomic identification at species level. How did you obtain it?

--- the final identification was obtained on sequence analysis of the internal transcribed spacer (ITS) and D1/D2 ribosomal DNA regions of the isolate. 

Reviewer 2 Report

A report case of a COVID19 associated mucuromycosis is presented in some detail. However some aspects should be clarified:

line 70:  Any idea of what was the infection source in this patient? Dialysis tubing? respiratory intubation? airborne spores? It is clear that the infection occurred at the hospital

Line 89: the identification was based only in morphoplogical characters?

Line 90: it is understood that susceptibility to LAmB, posaconazole and isavuconalzole was detected at HD13 and treatment with LAmB was empirically started at HD3, Why it did not prevented the fungus to keep on growing? Why posaconazole or isavuconalzole were not administered to the patient?

Line 99: How did the fungus was identified to the species level, Only by morphological characteristics? Any molecular tests were performed?

lines 120-121:  Then it seems that corticoid and IL-6 treatments are responsible for the susceptibility to mucuromycosis, not SARS-Cov-2. As mentioned in the introduction, other mucuromycoses cases not associated with COVID19 were also presented by patients that were immunocompromised by corticoids, so does SARS-Cov-2 really plays a role in the infection by mucuromycetes?

Author Response

line 70:  Any idea of what was the infection source in this patient? Dialysis tubing? respiratory intubation? airborne spores? It is clear that the infection occurred at the hospital

--- The airborne spores seem likely the source, although the place where the patient inhaled is unclear. It is possible that he might have acquired them at his home where he was around multiple sources of mucormycosis spores (soil, organic matter). Absence of similar cases at the hospital in last 1 year since the COVID-19 cases started getting admitted to the hospital suggest that hospital was less likely to be the place where he acquired COVID-19. 

Line 89: the identification was based only in morphoplogical characters?

-- The identification was suspected initially on morphology consisting of unbranching sporangiophores, pauci-septate hyphae, round sporangiospores. It was confirmed to genus level by MALDI-TOF (at Mayo Clinic Reference Lab) and to species level on sequence analysis of the internal transcribed spacer (ITS) and D1/D2 ribosomal DNA regions (at CDC Mycotic Diseases Branch  Lab). 

Line 90: it is understood that susceptibility to LAmB, posaconazole and isavuconalzole was detected at HD13 and treatment with LAmB was empirically started at HD3, Why it did not prevented the fungus to keep on growing? Why posaconazole or isavuconalzole were not administered to the patient?

--- The patient was located at a community hospital in rural Delaware. Due to logistics and lack of sequence-based identification of fungi and susceptibility testing for fungal organisms, the isolate was sent to Mayo Clinic initially. Since Mayo clinic could not identify the species, it was re-routed to CDC lab. Isavuconazole is not available at the hospital pharmacy. Posaconazole was not used per hospital antibiotic guidelines where Liposomal -Amphotericin-B is the empirical antifungal. 

Line 99: How did the fungus was identified to the species level, Only by morphological characteristics? Any molecular tests were performed?

--- The fungus was identified to species level using sequence analysis of the internal transcribed spacer (ITS) and D1/D2 ribosomal DNA regions of the isolate at the Centers for Disease Control and Prevention.

lines 120-121:  Then it seems that corticoid and IL-6 treatments are responsible for the susceptibility to mucuromycosis, not SARS-Cov-2. As mentioned in the introduction, other mucuromycoses cases not associated with COVID19 were also presented by patients that were immunocompromised by corticoids, so does SARS-Cov-2 really plays a role in the infection by mucuromycetes?

-- The role played by long duration of treatment with corticosteroids and IL-6 inhibitors in suppressing the immune system is well-established from a pathophysiology standpoint  and these agents have been previously associated with invasive fungal infections. In this case, a life-threatening fungal infection occurred quickly and in the setting of COVID-19 which has been reported to cause lymphopenia similar to corticosteroids and IL-6 inhibitors. Quick occurrence of mucormycosis with a short course of IV corticosteroids and a single dose of IL-6 inhibitor raise concerns that COVID-19 induced lymphopenia might have played a role in the infection by mucormycetes.

Round 2

Reviewer 1 Report

Many thanks for your reply.

I recommend to accept this manuscript for publication in the current form.

Reviewer

Reviewer 2 Report

Now the questions are clear to me Thank you for your answers